# Unraveling the Dynamics of Estrogen and Progesterone Signaling in the Endometrium: An Overview

**DOI:** 10.3390/cells13151236

**Published:** 2024-07-23

**Authors:** Isabelle Dias Da Silva, Vincent Wuidar, Manon Zielonka, Christel Pequeux

**Affiliations:** Tumors and Development, Estrogen-Sensitive Tissues and Cancer Team, GIGA-Cancer, Laboratory of Biology, University of Liège, 4000 Liège, Belgium; i.diasdasilva@uliege.be (I.D.D.S.); vincent.wuidar@uliege.be (V.W.); manon.zielonka@uliege.be (M.Z.)

**Keywords:** endometrium, estrogen receptor, progestogen, progesterone receptor, transcription factor, signaling pathway

## Abstract

The endometrium is crucial for the perpetuation of human species. It is a complex and dynamic tissue lining the inner wall of the uterus, regulated throughout a woman’s life based on estrogen and progesterone fluctuations. During each menstrual cycle, this multicellular tissue undergoes cyclical changes, including regeneration, differentiation in order to allow egg implantation and embryo development, or shedding of the functional layer in the absence of pregnancy. The biology of the endometrium relies on paracrine interactions between epithelial and stromal cells involving complex signaling pathways that are modulated by the variations of estrogen and progesterone levels across the menstrual cycle. Understanding the complexity of estrogen and progesterone receptor signaling will help elucidate the mechanisms underlying normal reproductive physiology and provide fundamental knowledge contributing to a better understanding of the consequences of hormonal imbalances on gynecological conditions and tumorigenesis. In this narrative review, we delve into the physiology of the endometrium, encompassing the complex signaling pathways of estrogen and progesterone.

## 1. Introduction

Throughout their lifespan, women experience multiple hormonal fluctuations that can significantly impact their overall health and quality of life [1,2]. Estrogens and progesterone (P4) play an essential role in the growth and development of reproductive organs during puberty [3]. Furthermore, these hormones are crucial in maintaining the physiological rhythm of women by regulating key functions such as the menstrual cycle, fertility, and pregnancy [4,5,6,7]. However, the natural aging process results in the permanent cessation of estrogen secretion by the ovaries, leading to the end of the reproductive life and the onset of menopause (Figure 1).

The human endometrium plays a vital role in ensuring the perpetuation of the human species. The endometrium is a complex multicellular tissue lining the inner wall of the uterus where fertilized eggs implant and develop into embryos during pregnancy. It is a highly dynamic entity crucial in establishing and sustaining pregnancy. It also provides vital nutritional support to the embryo until the placenta ensures this function throughout the pregnancy [10,11]. Therefore, extensive research in the field of medically assisted reproduction has been carried out to provide an alternative bio-artificial device [12]. In this context, the development of artificial wombs could provide to extremely premature infants an environment to nurture their development [13]. Despite progress in 3D organoid culture, it is currently not possible to develop fully functional human endometrium ex vivo [14,15,16,17,18]. Nevertheless, these human endometrium organoids are crucial tools to study the complexity of the endometrium biology.

Imbalances or disruptions in estrogen and progesterone signaling can lead to various gynecological conditions related to the endometrium, affecting fertility and reproductive health. Endometriosis affects around 10% of women of reproductive age. It is characterized by endometrial-like tissue growing outside the uterus, leading to pain, inflammation, and potential infertility [19]. Another condition is endometrial hyperplasia, characterized by an overgrowth of the epithelial cell lining, which can result in abnormal bleeding and, in some cases, may progress to endometrial cancer [20]. Adenomyosis has more recently gained attention. It involves the abnormal presence of endometrial tissue within the muscular wall of the uterus, causing pain and heavy menstrual bleeding [21,22]. Furthermore, endometrial polyps and fibroids, benign growths within the endometrial layer or the uterine wall, respectively, can also lead to abnormal bleeding and fertility issues [23,24]. Studying estrogen and progesterone pathways will help construct a basis for the management of these conditions, identifying potential targets.

The endometrium undergoes cyclical changes, including regeneration, differentiation, and shedding of the functional layer during each menstrual cycle [25]. These processes are modulated by estrogen and progesterone, which are governed by the hypothalamic–pituitary–ovarian (HPO) axis throughout a woman’s reproductive life [26]. Nevertheless, the endometrium is also a matter of concern when menopause occurs, as menopause results from the ovaries ceasing to produce estrogen. Menopause is not a disease, but it is associated with several symptoms that highly impact the quality of life of women [27]. To alleviate these symptoms, menopausal hormone therapies (MHT), whether based on estrogen only or on a combination of an estrogen with a progestogen (combined MHT), remain the most efficient and the most frequently used treatment [28,29,30,31]. Thus, understanding estrogen and progesterone signaling in the endometrium is essential to optimize menopause hormone therapies, minimizing risks and maximizing benefits.

To face the various medical challenges affecting the endometrium, an in-depth knowledge of the impact of molecular mechanisms triggered by hormonal exposure on the endometrium is necessary. This narrative review covers the endometrial physiology, followed by a detailed examination of the molecular mechanisms associated with estrogen (ER) and progesterone receptor (PR) signaling.

## 2. Endometrial Physiology

### 2.1. Endometrium Biology

Structurally and functionally, the mucosal tissue is organized into two mains layers: the basal layer (stratum basalis) and the functional layer (stratum functionalis) (Figure 2) [32,33]. The basal layer is in direct contact with the myometrium and remains relatively stable across the menstrual cycle, hosting vital endometrial progenitor/stem cells essential for regenerating the functional layer after each menstrual cycle or in case of endometrial ablation [34,35,36,37,38,39]. However, a great deal of evidence suggests the existence of endometrial progenitor/stem cells in the functional layer as well [40,41,42,43]. In contrast to the basal layer, the functional layer, located at the uppermost part of the endometrium, is a dynamic germinal compartment comprised of the glandular and luminal epithelium resting on the basal laminal and a thick stroma enriched in matrix extracellular cells, stromal fibroblastic cells, blood vessels, and immunocompetent cells [44]. This layer undergoes numerous cyclical morphological and functional changes in response to ovarian hormones [14,45] and plays an essential role in embryo implantation and pregnancy development. Remarkably, the human endometrium exhibits a highly regenerative capacity, experiencing approximatively 450 cycles of shedding and regeneration throughout a woman’s lifetime but also during postmenopause under a proper MHT [11,46,47,48]. This physiological process of shedding and regeneration is a distinctive trait shared by both humans and higher-order primates, strongly implying the presence of adult stem cells [49,50].

### 2.2. Endometrial Cellular Diversity

Exploring the intricate functional landscape of the endometrium reveals a rich diversity of cell types derived from stem cells, characterized by their capacity for self-renewal and differentiation into multiple cellular lineages [51]. Several studies have identified OCT-4, NANOG, SOX2, and SOX9 as being specific markers of adult stem cells that are gradually lost during differentiation [52,53,54,55]. Additionally, the side population technique has demonstrated the independent generation of the two main epithelial and stromal lineages of the functional layer [56]. Currently, three main types of stem cells have been identified in the human endometrium [57]: (1) epithelial stem cells [58,59,60], (2) mesenchymal/stromal stem cells [61,62,63], and (3) perivascular stem cells [64]. Epithelial stem cells primarily reside in the basal layer of the endometrium. These cells are regulated through direct interactions with endocrine and paracrine factors, involving intricate connections that require hormones and/or growth factors from surrounding immune and stromal cells [65]. Under normal physiological conditions, 17β-estradiol (E2), the main estrogen, is known to induce the proliferation of epithelial cells. Studies on stage-specific embryonic antigen-1 (SSEA-1)-positive cells have revealed progenitor cell properties [66], showing that when exposed to E2, these cells can differentiate into mature epithelial cells [67,68]. This pattern is consistent with the high activity of the basal layer cells during the proliferative phase of the menstrual cycle, when estrogens predominate [55,69]. Mesenchymal stem cells are presumed to reside in the perivascular region of stromal layer, contributing to support of the architecture of blood vessels and endometrial glands [70]. Moreover, they play a role in regulating the immune responses of the endometrium [71,72,73]. These cells have been identified as having the ability to differentiate into endothelial, fibroblastic, and smooth muscle cells [74]. Furthermore, perivascular stem cells are presumed to reside in the perivascular region, where they actively contribute to the establishment and regulation of the vascular system within the endometrium. These stem cells express specific markers such as CD146, PDGFRβ, and SUSD2 and have the ability to generate stromal and epithelial cells of the endometrium [60,62,75,76,77,78,79,80].

Current cutting-edge techniques such as single-cell RNA sequencing (sc-RNA-seq) provide high resolution for studying and characterizing individual cells rather than gathering them within a cell population [57]. These methods offer to researchers the opportunity to unveil rare or heterogeneous cell populations that may be overlooked by whole-tissue analysis [81]. In this context, Garcia-Alonso and colleagues [82], employing an integrated approach combining sc-RNA-seq with spatial transcriptomics, precisely highlighted cellular heterogeneity in the endometrium across different phases of the menstrual cycle. Using endometrial tissues from proliferative and secretory phases of 15 premenopausal patients, they identified 14 clusters of distinct cell populations in the endometrium, which can be categorized into five specific groups: (1) immune cells (myeloid and lymphoid), (2) stromal cells (fibroblasts (C7+)), (3) perivascular cells and smooth muscle cells (PV STEAP4 and PV MYH11), (4) endothelial cells (venous and arterial), and (5) epithelial cells. Furthermore, these analyses uncovered that epithelial cells are segmented into four subpopulations, defined by their unique marker expressions: (1) SOX9^+^ epithelial cells, which include SOX9^+^/LGR5^+^ cells located in the surface epithelium and SOX9^+^/LGR5^-^ cells in basal glands; (2) ciliated epithelial cells (PIFO, TPPP3, and FOXJ1); (3) luminal epithelial cells (LGR5), and (4) glandular epithelial cells (SCGB2A2 and FOXA2). 

Employing sc-RNA-seq profiling and unbiased clustering of full-thickness human uterine tissues from both proliferative and secretory phases of the menstrual cycle, Wu and colleagues [83] expanded the endometrium cell characterization to stromal cells and their subpopulations. In this study, endothelial cells were identified as either inflammatory or secretory, while vascular smooth muscle cells were categorized into four sub-groups: (1) ADIRF^+^ vascular, (2) secretory, (3) inflammatory, and (4) DES^+^ cells. In addition, stromal cells were divided into four distinct subpopulations: (1) secretory stromal cells (SCGB1D2^+^), (2) SFRP4^+^ stromal cells, (3) DCN^+^ stromal cells, and (4) inflammatory stromal cells (IL6^+^). Notably, the SFRP4^+^ stromal cells promote the regeneration of endometrial epithelial glands and full-thickness endometrial lesion through the IGF-1 signaling pathway in vivo. This study highlighted the heterogeneity of stromal cells and identified previously overlooked subpopulations that are important during the menstrual cycle.

### 2.3. Menstrual Cycle and Preparation of Embryo Implantation

Monthly, the female reproductive system undergoes profound transformations through endocrine orchestration directed by the HPO axis. This intricate process, commonly known as the menstrual cycle, emerges from coordinated interactions among hormonal fluctuations—primarily estrogens (especially E2) and P4—intertwined with underlying biological transformations [5,84]. A normal menstrual cycle, averaging 28 days (with variations from 25 to 35 days in the majority of cases), prepares the female body for the possibility of pregnancy through three distinct phases: the proliferative phase, the ovulatory phase, and the secretory phase.

During the proliferative or follicular phase (day 1 to 14), estrogen signaling dominates [11]. Under the influence of follicle-stimulating hormone (FSH) secreted by the pituitary gland, ovarian follicles develop and secrete estrogens. E2 plasma concentration fluctuates between 31–90 pg/mL and could reach 500 pg/mL just before the ovulatory phase, typically around the 14th day of the menstrual cycle [85]. The rise in estrogen levels promotes the proliferation of epithelial and stromal cells in the endometrium, contributing to its reconstruction [32,33]. Notably, the proliferation of the epithelium results from a paracrine signaling initiated by the stromal cells [86]. Moreover, cilium differentiation is engaged, glands become straight and narrow, and the stroma thickens and vascularizes. This hormonal surge also stimulates the production of cervical mucus, facilitating the transit of spermatozoa [87,88,89].

During the secretory or luteal phase (day 14 to 28), the ruptured follicle degenerates (follicular atresia) and transforms into an endocrine gland, the corpus luteum, under the influence of hormonal impregnation [90,91]. This gland secretes significant amounts of P4 (more than 10 ng/mL) [85,92], initiating a cascade of events crucial for the prospect of a future pregnancy, with levels reaching 150 ng/mL at term [92]. P4 induces specific modifications in the endometrium, such as the twisting of endometrial glands [93], thickening of cervical mucus, increased growth of spiral arteries [94], and a slight elevation in body temperature. Concurrently, the surrounding arteries and blood vessels lengthen up to the superficial layers and increase in diameter to nourish the potential embryo [95,96]. One of the critical processes during this phase is the decidualization of endometrial stromal cells, transforming them into specialized secretory decidual cells [97]. This process is finely regulated by an increase in P4 and cyclic adenosine monophosphate (cAMP) signaling, coupled with low levels of E2 [98,99,100]. It entails profound changes in both the morphology and function of stromal cells. Indeed, these cells undergo enlargement, adopting a rounder shape and experiencing an increased cytoplasmic volume. Decidualized cells become specialized and produce various P4-dependent factors essential for the establishment and maintenance of pregnancy. Among these factors are prolactin (PRL), glycogen, tissue factor, insulin-like growth factor-binding protein 1 (IGFBP-1), and the transcription factor Forkhead box O1 (FOXO1), contributing to the transformation of the endometrium towards a receptive state [37,101,102,103,104,105,106,107]. Alterations in the decidualization of the endometrial stroma have been associated with recurrent miscarriages, infertility, and other clinical pathological conditions such as endometriosis and endometrial cancer [25,108,109,110,111,112].

These changes are all designed to create an optimal environment in case of fertilization, providing a conducive site for embryonic implantation. In a normal menstrual cycle of 28 days, the window of implantation emerges as a critical timeframe from the 20th to the 24th day of the secretory phase, approximately 6 to 10 days after ovulation. The endometrium, now receptive due to the precise synchronization of hormonal events and cellular transformations, provides the appropriate conditions for successful blastocyst implantation [37,113,114,115]. In the absence of implantation, the cessation of P4 production by the corpus luteum triggers arterial contraction, causing devascularization and subsequent cell death within the functional layer, leading to the desquamation of the superficial layer of the endometrium and the onset of menstruation [37,116]. This process signals the beginning of a new menstrual cycle.

## 3. ER and PR Signaling in the Endometrium

Recent advances in research models such as murine models and in vitro techniques such as 3D human endometrial tissue cultures known as “organoids” have greatly enhanced our understanding of the ER and PR signaling in the endometrium [16,82,117,118,119]. Murine models are essential for investigating interactions within the endometrial microenvironment and for analyzing dynamic changes in the endometrium such as the decidualization of stromal cells [120] and the impact of hormonal imbalances, particularly in the case of menopause. These complex aspects remain challenging to model in vitro. The emergence of 3D culture of human endometrial organoids and of organ-on-a-chip models, which faithfully reproduce the architecture and functions of the epithelial compartment of the endometrium, is crucial for deepening our understanding of cellular dynamics and hormonal signaling in human tissue [14,121,122,123]. These developments, coupled with cutting-edge technologies like spatial genomics, provide a much more precise view of the cellular and molecular mechanisms regulating endometrial functions [16,82]. This integrated approach is instrumental in developing more effective therapeutic strategies in the field of women’s health.

### 3.1. ER Signaling in the Endometrium

ER, primarily ERα encoded by the *ESR1* gene on chromosome 6 (6q25.1) and ERβ encoded by the *ESR2* gene on chromosome 14 (14q23.2), are key players in the development and function of the endometrium. Studies using mutant mouse models have been fundamental in elucidating the specific roles of these two nuclear receptors. Female mice knocked-out for ERα (ERα-KO) are infertile. They display endometrial development anomalies, such as a hypoplastic uterus, hyperemic ovaries with no detectable corpus luteum, reduced PR expression, and insensitivity to estrogens, shedding light on the importance of ERα in regulating the endometrium and fertility [124,125,126,127]. Concurrently, ERβ-KO female mice show anomalies in endometrial differentiation, being sub-fertile and mainly lacking effective ovulatory function but without differences in responsiveness to E2, confirming the modulatory role of ERβ in this process [128,129,130]. The endometrium physiology of mice with deletion of both receptors are similar to those who are ERα-KO [124,126,130,131]. Altogether, these studies highlight the central role of ERα in mediating the estrogenic action in the endometrium.

#### 3.1.1. ER Signaling: Genomic Pathway

ERα comprises several key functional domains, including the amino-terminal A/B domains containing the ligand-independent activation function AF-1 and the carboxy-terminal E/F domains containing the ligand-binding domain and the ligand-dependent activation function AF-2 [132,133]. These activation functions are involved in the transcriptional activity of ERα. Research on AF-1 and AF-2 mutant mice has provided invaluable insights about the role of these domains in ER functionality [134]. Mice lacking ERαAF-1 or ERαAF-2 are sterile [135,136,137]. Notably, in ERαAF2^0^ mice, the uterotrophic effect induced by E2 is entirely abrogated, indicative of the predominant role of AF-2 in uterine growth [136,137]. Arao and colleagues highlighted the dependency of AF-1 on AF-2, reaffirming the necessity of AF-2 function for the estrogenic responses [137]. However, Abot and colleagues evidenced that specific E2 responsive-genes involved in proliferation, such as the insulin-like growth factor (*IGF-1*), are tightly regulated by AF-1 [138]. Thus, both AF-1 and AF-2 of ERα play an essential role for the physiological function of the uterus.

In its inactive state, ERα is maintained in a ligand-binding competent state by stabilizing chaperone proteins. In the human endometrium, several heat-shock proteins (HSP), including HSP27, HSP60, HSP70, HSP90, and αβ crystallin (CryAB) [139,140,141,142], have been identified. Estrogen binding to ERα triggers its dissociation from chaperone proteins and its dimerization, a critical process for its nuclear translocation, allowing the interaction of ERα with estrogen-response elements (ERE) in DNA [143]. This interaction represents the classical mechanism through which ERα modulates gene expression, regulating the activation or repression of specific genes [144,145] and influencing cellular proliferation, differentiation, and other key endometrial processes such as uterine development, fertility, blastocyst implantation, and early embryonic stages [146,147]. Beyond its ability to bind directly to ERE, ERα can also regulate gene expression through interactions with other transcription factors. These interactions include factors such as activator protein-1 (AP-1), specificity protein-1 (SP-1), and nuclear factor kappa-B (NFκB) [148,149,150,151], thus allowing ERα to influence the expression of genes lacking traditional ERE sites [133,152]. In addition, recent work integrating *ESR1* cistromes and endometrial biopsy transcriptomes [153] revealed a preferential enrichment of HOX motifs [154] and bHLH motifs [155] in mid-secretory ERα peaks, highlighting the implication of these binding sites in estrogen signaling. These findings, along with the identification of other significant motifs, have deepened our understanding of the molecular mechanisms orchestrated by ERα signaling in the endometrium, revealing a more sophisticated transcriptional regulation than initially anticipated [153].

#### 3.1.2. ER Signaling: Non-Genomic Pathway

In addition to their classical genomic response, ERα exhibits a form localized in the cell membrane, termed membrane-associated ERα (mERα), accounting for 5% of ERα cell content [156]. Unlike cytoplasmic ERα that mediates long-term effects through transcriptional regulation, mERα is associated with rapid, non-genomic signaling known as “Membrane-Initiated Steroid Signaling” (MISS). One of the first observations of these rapid effects was reported in 1975 as an increase in intracellular calcium concentration [157] via PLCβ activation in rat osteoblasts [158]. Further studies emphasized that this rapid signaling is associated with the activation of Gα and Gβγ proteins [159], the regulation of potassium channels, the activation of the ERK/MAPK pathway, and the regulation of lipid kinases such as phosphatidylinositol-3-kinase (PI3K) and adenylate cyclase, leading to the increase in cAMP levels in the uterus [160,161]. Numerous studies have been pivotal in elucidating this MISS signaling, notably through the generation of estrogen dendrimer conjugates (EDC), where E2 is covalently bound to polyamidoamine dendrimer macromolecules, preventing its translocation to the nucleus due to their size and charge [162]. In the uterus, the use of EDC activates ERK/MAPK signaling but does not enable the induction of epithelial cells growth or an adequate uterotrophic response in contrast to E2 [163]. In the C451A-ERα mutant mice, ERα is no longer palmitoylated and thus not anchored at the plasma membrane [164,165]. Similarly to wild-type mice, the uterus of C451A-ERα mice responds to E2, indicating that the MISS effects of ERα are not critical for uterine functions. However, the loss of membrane ERα leads to infertility due to ovarian defects, highlighting the involvement of ERα-MISS pathway in ovarian functions [164]. In another mER mutant mouse model, R264A-ERα mice, E2 is unable to trigger mERα-PI3K-Src complex formation and Gαi activation, highlighting that this mutation prevents proper MISS pathway signaling [166]. Interestingly, R264A-ERα mice are fertile, in contrast to C451A-ERα mice. However, both R264A-ERα and C451A-ERα mouse models have shown that mERα modulates the reactivity of the uterus to estrogen treatments since it increases the tissue sensitivity to physiological E2 [167] but decreases its response to supraphysiological E2 [166,167].

Besides ERα and ERβ, the role of the orphan G protein-coupled receptor (GPR30) in estrogen signaling has been highlighted, leading to its reclassification as a G protein-coupled estrogen receptor (GPER) [168,169,170]. GPER has cyclic expression levels in endometrium, with a significant increase during the proliferative phase in response to E2 and a decrease during the secretory and menstrual phases [171]. However, GPER-KO mice models exhibit no reproductive defects [172], minimizing the role of GPER in endometrium physiology.

#### 3.1.3. Key Mediators of ERα Signaling in the Endometrium

ERα signaling is mainly implicated in the proliferation of the epithelial cells of the endometrium occurring during the proliferative phase of the menstrual cycle and in the preparation to decidualization and proper receptivity during the secretory phase.

##### Proliferation-Related Mediators

During the proliferative phase, even though both epithelial and stromal cells express ERα, the proliferation of the epithelium compartment results from a paracrine signaling initiated by ERα expressed by stromal cells (Figure 3). The specific ablation of ERα in epithelial cells revealed that estrogen-induced proliferation is unexpectedly independent of epithelial ERα [173,174,175]. During the proliferative phase, the activation of stromal ERα stimulates the production of **IGF-1** that promotes the proliferation of the epithelial cells [176]. Moreover, ERα and ERβ differentially regulate the expression of IGF-1 mRNA, as ERα promotes its expression, while ERβ inhibits it in the endometrial stromal cells of mice [177]. This growth factor, acting via its receptor IGF-1R on epithelial cells, activates the PI3K/AKT/mTOR signaling pathway, which is essential for various cellular processes. AKT phosphorylates the glycogen synthase kinase 3β (GSK3β), promoting nuclear translocation and accumulation of cyclin D1, a crucial regulatory factor of the cell cycle. This mechanism stimulates the proliferation of epithelial cells, thus underlying the importance of the interactions between estrogens and intercellular signaling pathways in the regulation of the endometrium [86,178,179,180]. Interestingly, female IGF1-KO mice exhibit a hypoplastic uterus and an inability to ovulate, leading to infertility, which highlights the vital role of IGF-1 in uterine function and fertility [181,182]. However, it is important to note that IGF-1 alone cannot completely replace the effects of E2 on the proliferation of endometrial epithelial cells, as altering its secretion does not alter this process [180,183,184]. These observations sustain the involvement of other mediators to regulate epithelial cell proliferation.

The fibroblast growth factor (FGF) family and transcription factor CCAAT/enhancer-binding protein β (C/EBPβ) have been identified to contribute to the epithelial cell proliferation. The paracrine activity of **FGF** via their receptors (FGF-R) induces the proliferation of the luminal epithelium via the activation of ERK/MAPK and PI3K/AKT signaling cascades [155,185,186,187]. FGF-9 is a survival and mitogenic factor, predominantly secreted by stromal cells in response to E2 and prostaglandins (PGE) during the proliferative phase [188,189]. Research by Šućurović and colleagues [190] revealed the critical importance of FGF-9 in creating a favorable microenvironment for successful implantation and establishment of gestation in mice. Similarly, FGF-10 and BMP-8a have been identified as stromal-derived paracrine factors regulating the growth and controlling the proliferation of E2-dependent epithelial cells in mice [191].

The transcription factor **C/EBPβ** has been identified as a regulator of the proliferation and differentiation of epithelial and stromal cells [192,193]. Plante and colleagues reported a cyclic expression of C/EBPβ in the endometrium through menstrual cycle. Indeed, C/EBPβ is expressed by both epithelial and stromal cells during the proliferative phase, but its expression is increased in both cell types at the onset of the mid-secretory phase [192]. In addition, the administration of E2 to ovariectomized mice induces the expression of C/EBPβ in the uterine epithelium and stroma [194]. Indeed, C/EBPβ-KO mice exhibit a significant reduction in E2-induced epithelial cell proliferation compared to wild-type mice, indicating the implication of C/EBPβ in this process [194].

Both IGF-1 and C/EBPβ expression in response to E2 is dependent of ERα expression by stromal cells but not of epithelial ERα expression, reinforcing the importance of the paracrine regulation of epithelial cell proliferation [174,195]. Other E2-responsive genes, such as cyclin-dependent kinase inhibitor 1a (***CDKN1A***) and mitotic arrest deficient 2-like protein 1 (***MAD2L1***), have been identified and are known to play a role in the cell cycle regulation [174,195].

##### Mediators Involved in the Preparation to Decidualization and Implantation

During the menstrual cycle, E2 blood concentration follows a double-wave curve, reaching a maximal level in the proliferative phase but rising a second time during the secretory phase. This indicates a contribution of E2 to proper decidualization. Moreover, E2 has been found to regulate the expression of key mediators involved in the endometrial receptivity that are summarized hereafter (Figure 3).

Mucin-1 (**MUC1**), a transmembrane glycoprotein present in the uterine epithelium, plays a protective role by defending the endometrium against pathogens and foreign particles. MUC1 is sensitive to estrogens, and its expression is attenuated at the time of implantation to facilitate epithelial remodeling [196,197]. Indeed, high expression of MUC1 can inhibit embryonic adhesion, while a decrease in its expression, induced by P4 via the Hedgehog signaling (IHH/COUP-TFII) during the secretory phase, is essential for facilitating embryonic implantation [197,198,199,200,201]. During the window of implantation, fine regulation of MUC1 is crucial to allow the attachment of the embryo, representing one of the determining markers of uterine receptivity [202].

Leukemia inhibitory factor (**LIF**), a predominant pro-inflammatory cytokine in the glandular epithelium before implantation, is induced by E2 in the luminal epithelium in mice [5,203,204]. By binding to a receptor complex composed of LIF-R and Gp130, LIF activates the JAK/STAT3 pathway, which, in synergy with PR signaling, enhances uterine receptivity to the embryo, thereby promoting the success of implantation [204,205,206,207,208]. LIF, also expressed in the stroma surrounding the implanting blastocyst [209], is involved in many other signaling pathways, including IGF-1, VEGF/HIF-1α, TGFβ, FGF, Wnt/β-catenin, and NOTCH, demonstrating its versatility in endometrial regulation [205,206]. As a pleiotropic factor, LIF is essential for effective bidirectional communication between the endometrium and the embryo, the decidualization of stromal cells, and blastocyst development. Abnormalities in LIF expression, particularly abnormal low levels, are associated with embryonic implantation dysfunction, potentially leading to infertility in women [210]. In female LIF-KO mice, infertility results from disruption of embryo apposition to the endometrium and a defect in decidualization, hindering normal implantation and pregnancy development [211,212,213]. However, these defects can be compensated by the administration of exogenous LIF, replacing the effect of E2 production during the implantation and restoring a receptive uterus [204,206,214].

Liang and colleagues [215] highlighted the essential role of the early growth response 1 (**EGR1**) factor in endometrial receptivity. They reported that the estrogen induction of LIF results in STAT3 phosphorylation in mouse uterine stromal cells, which activates the multifunctional transcription factor EGR1. EGR1 is also quickly and transiently induced in stromal and epithelial cells by ERα-ERK1/2 axis in response to estrogens or endocrine disruptors with estrogenic activity, such as bisphenol A [216,217]. Furthermore, its expression is particularly marked in the subluminal stromal compartment of the murine endometrium surrounding the blastocyst during the window of implantation [215,216], but it decreases considerably after implantation. Female EGR1-KO mice exhibit implantation and decidualization problems, highlighting the crucial importance of EGR1 [217,218]. Further studies have reinforced the importance of EGR1 in decidualization. Indeed, mice with EGR1 deletion show persistent epithelial endometrial cell proliferation due to the inability of P4 to abolish E2-dependent proliferation in these mice. Moreover, the knock-down of EGR1 in pre-decidual human endometrial stromal (HES) cells leads to the inability of these cells to respond to decidualization signals [217,219]. Additionally, the work of Szwarc and colleagues demonstrated that disturbances in EGR1 levels are associated with endometrial pathologies such as recurrent implantation failures [219] and endometriosis [220]. Altogether, these data emphasize the importance of EGR1 transcriptional activity in endometrial receptivity via the LIF-STAT3 and ERK/MAPK pathways resulting from ERα activation in epithelial cells. Moreover, EGR1 also induces the expression of c-kit [221], WNT4 [215], and ADAMTS-1 [222], which are involved in the implantation process.

While ERα signaling orchestrates fundamental endometrial processes during the proliferative and the secretory phases of the menstrual cycle, the importance of PR signaling is also demonstrated during the secretory phase and the decidualization. The synergy between these two pathways is crucial for maintaining the physiological integrity and functionality of the endometrium, emphasizing the complex and regulated interaction between these two hormonal systems.

### 3.2. PR Signaling in the Endometrium

#### 3.2.1. Dynamics and Regulation of PR in the Endometrium

PR presents two major isoforms, PR-A and PR-B, transcribed from two alternative promoters located on the same gene (*PGR*) found on chromosome 11 (11q22-q23) [223,224,225]. Similarly to ERs, PRs are composed of a DNA-binding domain, a ligand-binding domain, a specific sequence for nuclear localization and dimerization, an inhibitory domain, and activation functions. The PR-A variant contains AF-1 and AF-2 functions, while PR-B is characterized by an additional N-terminal region of 164 amino acids, i.e., the “B-upstream segment” (BUS domain), conferring an additional AF-3 function [226,227]. The functional differences between PR-A and PR-B are often attributed to the PR-B-specific BUS domain.

During the proliferative phase, the expression of PRs is mainly ensured by estrogens acting via ERα in both stromal and epithelial cells. In response to P4, PR-A and PR-B often form homodimers or heterodimers, exerting specific transcriptional activities and targeting different sets of gene promoters. During the menstrual cycle, the response to P4 may differ based on cell-specific variations in the expression levels of PR-A and PR-B. During the proliferative phase, the expression of these receptors increases in parallel with estrogen blood levels. Conversely, at the end of the secretory phase, a decrease in PR-A is observed in epithelial cells, while the level of PR-B remains constant, contributing to glandular secretion. Meanwhile, stromal cells predominantly express PR-A throughout the menstrual cycle [228]. In these cells, P4/PR-A exerts a negative feedback on ERα signaling, thus modulating downstream hormonal effects, particularly by inhibiting active cell division and promoting cellular differentiation [229]. These processes are essential for decidualization and the success of implantation.

Kurita and colleagues showed that the regulation of epithelial PRs is influenced by paracrine mechanisms mediated by steroid receptors in the stroma, shedding light on the complex interconnection between stromal and epithelial compartments [230,231]. Even if both isoforms are expressed in endometrium, PR-A seems to be sufficient for normal uterine function and implantation [232,233,234]. Studies on female PRA-KO mice revealed their infertility is mainly due to implantation and decidualization defects, thus highlighting the importance of PR-A in these processes [234,235]. Moreover, the uterine wet weight is dependent on PR-A isoform since in PRA-KO mice, the uterine wet weight is not increased in response to E2 [234]. In addition, the use of PRA-KO mice revealed that PR-B contributes to the proliferation of epithelial cells in response to P4, whereas PR-A is involved in differentiation and proliferation arrest. These results emphasize opposite roles of PR-A and PR-B in controlling the proliferation of epithelial cell in the endometrium [229]. Therefore, in response to P4, PR-A isoform inhibits both the E2-induced proliferation and the P4/PR-B-induced proliferation [234].

The balance in the expression of PR-A and PR-B isoforms is crucial to maintain the homeostasis of epithelial cell proliferation and to prevent uterine anomalies. The work of Brosens and colleagues [236] revealed that the overexpression of both PR-A and PR-B receptors in primary HES inhibits decidualization, a phenomenon dependent on the activation of the protein kinase A (PKA) pathway regulated by intracellular cAMP levels. PKA activation contributes to sensitizing stromal cells to the effect of P4, notably via the induction of specific PR co-activators essential for embryonic implantation [102,236]. Moreover, the overexpression of PR-A leads to a hyperproliferative state in the luminal and glandular endometrium and to infertility [237,238].

#### 3.2.2. PR Signaling: Genomic Pathway

In the classical genomic pathway, cytoplasmic PRs remain inactive in association with HSP proteins until P4 binds to their LBD, causing their dimerization and nuclear translocation. Once in the nucleus, these activated PRs specifically bind to palindromic DNA binding sites called progesterone response elements (PRE), leading to their recruitment on promoters or enhancer elements as well as the recruitment of co-regulators, thus modulating gene expression downstream of PR signaling [239,240,241]. Among the most well-characterized co-regulators of PR signaling, the p160/SRC co-regulator family including SRC-1, SRC-2, and SRC-3 modulates PR transcriptional activity, thus contributing significantly to uterine function [242]. SRC-1 and SRC-2, both expressed in the epithelium and the stroma of the endometrium, appear to be essential for decidualization and uterine response to steroid hormones. While SRC-3 is less present in the pre-implantation uterus, it was identified by Maurya and colleagues as important for stromal decidualization in vitro as well [243]. Similarly to ER, PR can also indirectly modulate gene expression by directly interacting with DNA-bound transcription factors such as *AP-1*, *SP-1*, *NFκB,* and *STAT3* [244].

#### 3.2.3. PR Signaling: Non-Genomic Pathway

PR can also be localized at the plasma membrane through its interaction with G protein-coupled receptors and specific transmembrane receptors, triggering rapid non-genomic effects [245,246,247]. In contrast to mERα pathway, the PR non-genomic signaling is mediated by specialized receptors that are completely different from PR nuclear receptors. Indeed, PRs localized at the plasma membrane are divided into two main categories: (1) the membrane progesterone receptors (mPRs) including five isoforms that are members of class 2 of the progesterone and adipoQ receptor (PAQR) family and (2) the progesterone membrane components (PGRMC) family including PGRMC1 and PGRMC2 [248,249,250,251,252,253]. Although the involvement of these receptors in PR signaling is gaining more interest [250,254,255,256], the rapid effects of P4 through the membrane pathway remain poorly documented in the endometrium and need further research to elucidate their specific role [257].

#### 3.2.4. Key Mediators of PR Signaling in the Endometrium

P4/PR signaling inhibits the proliferative effect of E2 observed in epithelial cells and controls the decidualization of stromal cells. In addition, P4/PR signaling drives the differentiation of luminal epithelial cells into ciliated cells and of glandular epithelial cells into secretory cells. Numerous mediators and molecular modulators, essential for decidualization and receptivity, depend on P4/PR signaling and are involved in epithelial–stromal interactions [258] (Figure 4).

##### Mediators Counteracting ER Signaling

To promote decidualization and implantation, P4 needs to antagonize the proliferation of epithelial cells induced by E2. To do so, P4 induces the expression of IGFBP-1 that can segregate IGF-1. It leads to the reduction in the activation of AKT in epithelial cells, thus reducing the inactivating phosphorylation of GSK3β, resulting in the loss of nuclear cyclin D1 and subsequent proliferation [86,259]. On the other hand, P4 can also inhibit the E2-induced epithelial cell proliferation through epithelial EGR1 expression. Indeed, epithelial cells knocked-down for EGR1 exhibit a defect of P4 proliferative inhibitory action, highlighting the role of epithelial EGR1 in P4 regulation of proliferation [217].

Some P4/PR-induced proteins can also counteract estrogen signaling to promote decidualization. Indeed, Kurihara and colleagues revealed that P4 induces stromal **COUP-TFII** expression [199], resulting in the reduction in the expression of SRC-1 and ERα in the epithelial compartment. Moreover, COUP-TFII-deficient mice exhibit embryonic implantation dysfunction and placental malformation [260]. Hence, by this mechanism, P4 counteracts epithelial estrogenic activity, thus initiating epithelium differentiation and favoring embryo implantation [199,200].

Several studies have highlighted the role of the basic helix–loop–helix transcription factor (**HAND2**) in PR signaling. HAND2 acts as an anti-proliferative mediator on endometrial epithelial cells by inhibiting the mitogenic activity of FGF via the ERK/MAPK and PI3K/AKT signaling pathways [185,261], again highlighting the importance of PR-A in regulating endometrial differentiation and decidualization [155]. Oh and colleagues revealed that COUP-TFII binds to a genomic region near the *HAND2* gene and is characterized by active histone marks, enhancing HAND2 expression in preparation for embryonic implantation in the endometrium [262].

Some regulators of ER signaling are also found in PR signaling, such as **STAT3**, which can be activated by a variety of cytokines, including LIF [204]. STAT3 specifically interacts with PR-A, the main isoform found in the endometrium to modulate uterine function and, more especially, preparation for embryonic implantation for successful pregnancy [208]. The cistrome analysis of endometrial stromal cells revealed that P4/PR interaction rapidly induces the promyelocytic leukemia zinc finger protein (**PLZF**), another PR signaling mediator, which acts as a transcriptional repressor of E2-induced EGR1 previously induced by E2 to allow the P4-induced decidualization of HES cells during the secretory phase of the menstrual cycle [263,264]. The importance of PLZF was confirmed in HES cells treated with RU-486 and in PR-KO mice, where a significant decrease in PLZF expression was observed [264].

##### Main Mediators of Decidualization

Studies using high-density DNA microarray analysis combined with PR-KO mice highlighted the Hedgehog pathway as a downstream target of P4 [198,265]. Especially, **IHH** is induced in response to P4 in the luminal and glandular uterine epithelium at the time of implantation [265,266,267] and acts on adjacent stromal cells [201]. The subsequent activation of the Hedgehog signaling pathway by IHH triggers decidualization of stromal cells, preparing the endometrium for embryonic implantation [268]. Luminal epithelium-secreted IHH initiates a complex interaction with stromal cells by binding to its receptor patched-1 (PTCH1), inducing a conformational change of smoothened (SMO) protein that results in its activation [269]. This SMO activation triggers an intracellular signaling cascade involving the transcription factors GLI1 and COUP-TFII (chicken ovalbumin upstream promoter transcription factor II, also known as NR2F2) in the uterine stroma [270,271].

The orphan nuclear receptor **COUP-TFII**, which is spatiotemporally regulated throughout the menstrual cycle, is initially detectable during the proliferative phase, decreases at the beginning of the secretory phase in the functional layer, then progressively reappears in the basal layer as the secretory phase progresses. This dynamic expression coincides with post-menstrual regeneration of the endometrium following variations in steroid hormones during the cycle [272]. COUP-TFII induces the expression of bone morphogenetic protein 2 (BMP2), a key mediator of stromal cell decidualization [273,274,275,276,277].

**BMP2**, a morphogen belonging to the TGFβ superfamily, is expressed in the uterine stroma near the implantation site. Its expression, stimulated by COUP-TFII, is maintained during the early phase of decidualization and progressively surround the implanted embryo, highlighting its essential role in creating an environment conducive to pregnancy [273,274,275,276,277]. This regulation is crucial since the absence of COUP-TFII in the murine stroma leads to the failure of decidualization, a defect that can be compensated by adding BMP2 to the uterine horn [199]. BMP2 induces the SMAD pathway through the activation of the pairing of ALK3 type I receptor and BMP type II receptor. These phosphorylated SMAD then form a complex with SMAD4, subsequently translocated into the nucleus to regulate the expression of target genes. One of the important effects of this signaling cascade is the negative regulation of the expression of intercellular adhesion molecule 1 (ICAM-1) via the upregulation of the DNA-binding protein inhibitor 3 (ID3), which promotes a uterine environment conducive to implantation and the maintenance of pregnancy [278,279,280].

Through the SMAD pathway, BMP2 induces **WNT4** expression in stromal cells that activates the canonical Wnt/β-catenin pathway necessary for the transcriptional regulation of factors initiating decidualization and embryo attachment [281,282,283,284,285]. A deficiency in stromal β-catenin fails to upregulate IHH in the epithelium as well as GLI1 and PTCH1 in the stroma, suggesting that this pathway allows P4 to coordinate events in the underlying stromal and epithelial compartments for future implantation [286]. Moreover, WNT4 overexpression has been associated with a strong induction of IGFBP-1 and PRL mRNA levels, two drivers of decidualization that are inhibited by Dickkopf (DKK1), a well-characterized inhibitor of the canonical Wnt/β-catenin pathway [287]. To note, DKK1 is upregulated in decidualized stromal cells surrounding the gland [82].

Besides their role in the stroma, genes of the Wnt pathway are also expressed in the epithelial lumen of the endometrium, and Wnt target genes are evidenced in the ciliated epithelium [82]. Through Wnt signaling, P4 sustains the differentiation of the luminal epithelial cells into ciliated cells, a process initiated by E2 and requiring NOTCH inhibition [82,288]. While Wnt signaling contributes to cilium differentiation, NOTCH signaling is required for the differentiation of the glandular epithelium into secretory cells. This complex regulation highlights a cooperative interaction between the Wnt and NOTCH signaling pathways, preparing the endometrium for potential embryonic implantation through E2 and P4 [82].

Downstream of the BMP2-WNT4 signaling, the induction of the transcription factor **FOXO1** (Forkhead box O1) occurs in response to cAMP [285,289,290,291,292,293]. *FOXO1* has been identified as one of the first genes induced in endometrial stromal cells during human decidualization [294]. During the proliferative phase, FOXO1 is expressed in the cytoplasm of endometrial cells and moves to the nucleus during the secretory phase [295,296]. FOXO1 then induces the expression of IGFBP-1 that can prevent the proliferation-induced effects of epithelial cells, favoring stromal cell differentiation [297,298]. Labied and colleagues [292] also revealed that the cAMP-dependent expression of FOXO1 is amplified by the addition of P4 but not induced by P4 alone, highlighting the importance of the interaction between cAMP and P4 in the regulation of FOXO1. However, FOXO1, in response to cAMP signaling, can interact with PR even in the absence of its ligand P4, triggering the transcription of target genes such as *IGFBP-1*, *PRL,* and *WNT4*. Thus, P4 acts as a potentiator of these effects. This indicates an important synergy between FOXO1 and P4-bound PR in the regulation of gene expression in the endometrium [289]. Moreover, the overexpression of FOXO1 in human endometrial stromal cells promotes the expression of decidualization markers independently of steroid hormones and cAMP [289,290,299,300]. Besides its regulation by P4 and cAMP, FOXO1 can also be regulated through other signaling pathways. Indeed, FOXO1 is the target of PI3K/AKT pathway and can be phosphorylated at three sites (Thr-24, Ser-256, and Ser-319). When FOXO1 is phosphorylated by AKT, cytoplasmic retention is noted, inhibiting its transcriptional activity and thus reducing IGFBP-1 and PRL expression [301]. Moreover, the NOTCH pathway, specifically NOTCH-1, may also play a role in initiating decidualization by directly or indirectly regulating FOXO1 [302,303,304,305].

##### Other Partners of Decidualization

Several proteins have been identified to regulate the activation of the main decidualization axis. These mediators are also modulated by P4-PR signaling (Figure 4).

As aforementioned in the section dedicated to estrogen-dependent mediators regulating epithelial cell proliferation, P4 can also regulate the expression of C/EBPβ to control decidualization [306]. Indeed, P4 administration to ovariectomized mice induces a strong expression of C/EBPβ in both epithelial and stromal cells [194]. In addition, C/EBPβ expression is crucial to mediate decidualization since C/EBPβ-KO mice have a defect in stromal cell proliferation and differentiation [194]. Moreover, in stromal cells, C/EBPβ inhibits the activity of the p53 protein and stabilizes G2/M-inducing factors such as cyclin B2 and CDK1, highlighting its role in the proliferation of stromal cells in favor of decidualization [307]. The importance of C/EBPβ is further emphasized by the discovery that its regulation in stromal cells is directly influenced by the alpha receptor of retinoic acid (RARα), a crucial mechanism for decidualization and, consequently, for embryonic implantation [308]. Preliminary studies already revealed that female mice without the *C/EBPβ* gene are infertile, leading to implantation failure and a uterus unable to decidualize in response to a stimulus, suggesting a major role of this gene in the receptivity of the endometrium to embryonic implantation [194].

The protein **GATA2**, found in the luminal epithelium of the endometrium, also appears essential in the endometrium since mice deficient in GATA2 are infertile due to a failure of implantation and inadequate decidualization of the endometrium. This could be explained by a failure in the inhibition of ER signaling and a reduction in P4 target genes (only 3%) [309]. In addition, the absence of GATA2 significantly attenuates the expression of decidual markers in endometrial stromal cells such as IGFBP-1 and PRL [310]. Moreover, studies have identified that there is a GATA2-PGR-SOX17 regulatory network in favor of female fertility [311,312,313,314]. This network regulates PR signaling in early pregnancy and coordinates the activation of the main decidualization pathways, including the canonical Wnt signaling [309,315]. Studies indicate that Wnt/β-catenin signaling induces the transcription factor **SOX17** expression [316,317]. The interaction between β-catenin and SOX17 enhances its transcriptional activity [318]. Furthermore, transcriptomic and cistromic analyses revealed that *IHH* was among the genes regulated by SOX17 [319]. A peak of SOX17 has been observed at the site of attachment of the embryo, highlighting its role in regulating the action of PR in the uterine epithelium [313,320]. This mechanism explains why mice deficient in SOX17 are hypofertile due to (1) an alteration of LIF and (2) a disruption of IHH signaling, which are responsible for a failure of embryonic implantation [312,319].

**HOXA10** is another key mediator of epithelial–stromal interactions in the endometrium, orchestrated by PR signaling. This transcription factor is expressed in the epithelial and stromal cells of the endometrium, with a peak in epithelial cells during the window of implantation of the secretory phase [321,322,323,324]. Studies have shown that female mice deficient in HOXA10 are sub-fertile due to decidualization and implantation defects; an altered stromal response to P4, including dysregulation of COX-2 receptors, IGFBP-1, and prostaglandins (EP3 and EP4) [325,326]; as well as a disruption of WNT4 signaling [327]. Beyond its role in decidualization, HOXA10 facilitates the emergence of a permissive environment supporting embryonic implantation. It stimulates the expression of integrins (αvβ3 and α4β1) in the epithelium, and it induces the formations of pinopodes, apical epithelial projections that will facilitate embryonic adhesion [328,329].

**Follistatin** (Fst), encoded by the *FLRG* gene, exhibits a distinct spatiotemporal dynamic in the endometrium. Fst is expressed throughout the different phases of the menstrual cycle at a low level in epithelial cells during the proliferative phase and then highly expressed in stromal cells during the secretory phase and early pregnancy. Fst is strongly induced during in vitro decidualization of endometrial stromal cells in response to combined E2 and P4 treatments but not by individual treatments, again highlighting the importance of both E2 and P4 for the decidualization process [330]. Fst is essential for uterine function, as female Fst-KO mice are infertile or severely sub-fertile and characterized by defective decidualization with reduced stromal proliferation and differentiation. Fst is an effective chemoattractant protein inducing the migration of decidualized endometrial stromal cells via the activation of JNK signaling, in favor of successful pregnancy [331]. Furthermore, studies have revealed that when Fst is expressed, it suppresses the activity of activin B and the expression of inhibin βB in early pregnancy, two factors known to inhibit BMP signaling. By removing this inhibition, Fst promotes BMP2 signaling that converges with PR signaling to enable BMP-P4-regulated genes to be induced in favor of a receptive uterus. Without Fst, activin B is upregulated, leading to a failure of uterine receptivity [332,333,334].

Recently, Zhou and colleagues revealed that PR- and cAMP-responsive element-binding protein 1 (CREB1) directly binds to the promoter of *PTPN11*, encoding for the Src homology-2 domain-containing protein tyrosine phosphatase-2 (**SHP2),** thereby regulating its expression in response to decidualization signals (P4 and cAMP) in HES cells [335]. Hence, SHP2 also plays a crucial role in uterine receptivity. Indeed, SHP2 is expressed in a spatiotemporal manner in both epithelial and stromal cells during embryo implantation [336,337]. Functional studies have shown that female mice with SHP2 deletion are infertile due to the absence of embryo attachment to the uterine epithelium [337]. Additionally, the suppression of SHP2 in mouse stromal cells has led to sub-fertility, characterized by delayed embryo implantation. In vitro, the absence of SHP2 in HES cells leads to defective decidualization and differentiation blockade [336]. Several pathways are modulated by SHP2, including the ERK/MAPK signaling pathway, which is necessary for the nuclear translocation of PR and thus the transcription of PR-dependent genes that play a major role in decidualization [338].

Two immunophilin-binding proteins, FK506-binding protein 52 (**FKBP52**) and FKBP51, have been found to upregulate (FKBP52) or repress (FKBP51) the transcriptional activity of PR by binding the C-terminus end of the HSP90 protein, conferring co-chaperone activity with HSP90 that interacts with the ligand-binding domain of PR [339,340,341,342]. Specifically, FKBP52, a direct target downstream of HOXA10 [343], selectively upregulates the activity of PR-A in the uterus while having less impact on the activity of PR-B or ERα in this tissue [342]. Female mice deficient in FKBP52 are infertile due to a failure of implantation and decidualization [341,342], resulting in the reduction in PR transcriptional activity, leading to the reduction in the expression of many target genes involved in fertility [344].

## 4. Conclusions and Perspectives

The endometrium exhibits significant responsiveness to estrogens and progestogens, with extremely complex epithelial–stromal cell paracrine communications that are far from being fully understood. Indeed, these hormones regulate endometrium remodeling through cell proliferation and differentiation. Therefore, the understanding of the complex paracrine signaling is extremely important in better understanding the biology of the endometrium and identifying key pathways that could open new perspectives for the development of drugs that could prevent or treat medical issues such as fertility or menopause.

In this review, we highlighted that during the E2-dependent proliferative phase, key mediators including IGF-1, members of the FGF family, and the transcription factor CEBP/β orchestrate endometrial proliferation via the paracrine signaling pathways occurring between stromal and epithelial cells. Moreover, we emphasized the importance of the P4-dependent secretory phase in modulating a plethora of transcription factors, including FOXO1 and HOXA10. P4 also modulates the activation of signaling pathways, including the Hedgehog, NOTCH, and Wnt pathways, which are three important pathways that regulate stromal cell decidualization, a critical step essential for endometrial preparation and receptivity to potential egg implantation.

Interestingly, the lack of a model in which to study the human endometrium response at a molecular level is now filled with the emergence of organoids and organ-on-a-chip models that will allow the possibility of studying crucial steps such as endometrial proliferation, decidualization, the embryo–endometrial interface, and thus embryo implantation. These models will also provide the possibility to evaluate new drugs on human tissue before clinical trials.

## Figures and Tables

**Figure 1 cells-13-01236-f001:**
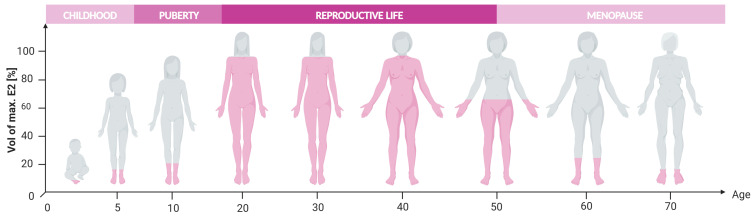
Estrogen levels throughout woman’s life, adapted from [8,9]. Created with BioRender.com (accessed on 16 June 2024).

**Figure 2 cells-13-01236-f002:**
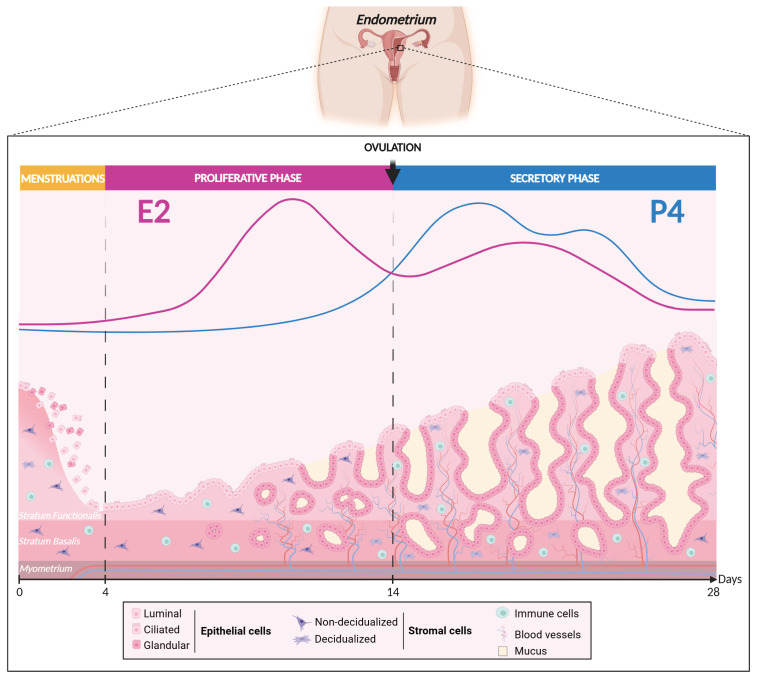
Woman’s menstrual cycle according to estrogen and progesterone fluctuation levels. Created with BioRender.com (accessed on 16 June 2024).

**Figure 3 cells-13-01236-f003:**
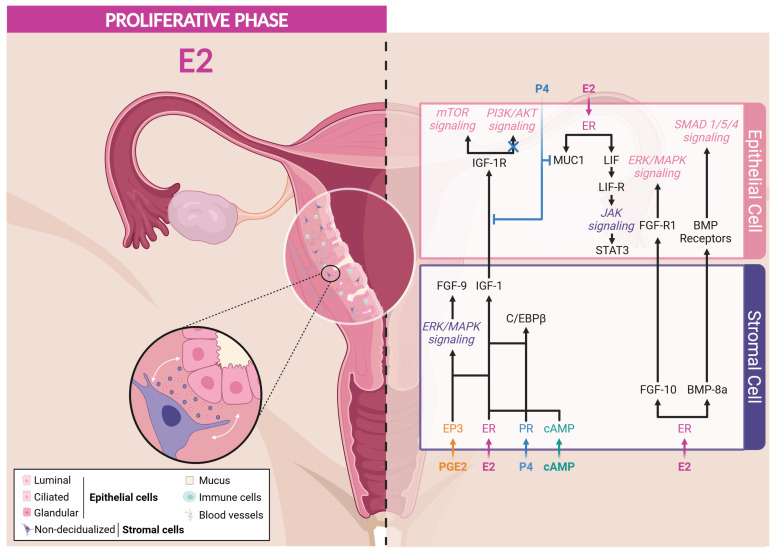
ER Signaling in response to estrogens during proliferative phase. Created with BioRender.com (accessed on 16 June 2024).

**Figure 4 cells-13-01236-f004:**
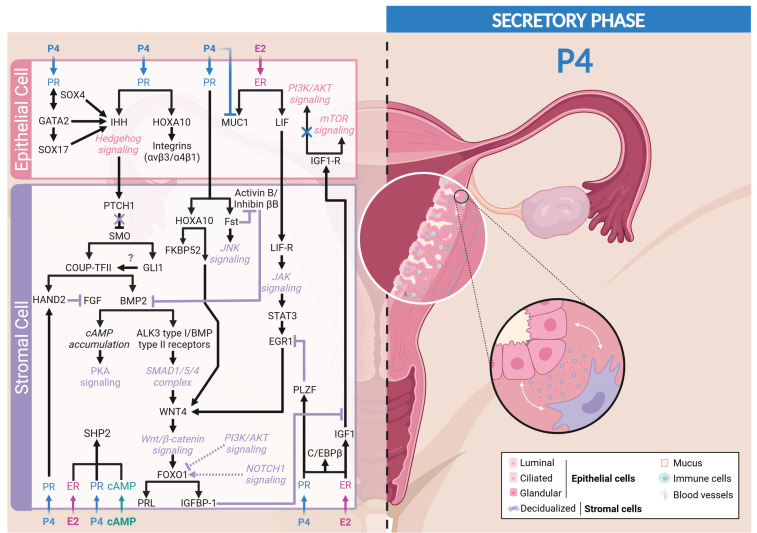
PR signaling in response to progesterone during secretory phase. Created with BioRender.com (accessed on 19 June 2024).

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
