# Peer review of "Unraveling the Dynamics of Estrogen and Progesterone Signaling in the Endometrium: An Overview"

_cells, 2024, doi:10.3390/cells13151236_

Round 1
Reviewer 1 Report (Previous Reviewer 1)
Comments and Suggestions for Authors This is a very good review that comprehensively introduces the pathways of action of estrogen and progesterone in the endometrium. The author cited a large number of references and summarized them very well. Progesterone promotes the transition of the endometrium from the proliferative phase to the secretory phase during pregnancy, provides oxygen, blood and other services to the uterus, increases the thickness of the endometrium and prepares for embryo implantation. In addition, progesterone can promote the development of mammary lobules and alveoli to prepare for postpartum breastfeeding. Progesterone can dynamically antagonize estrogen during pregnancy. In early pregnancy, it inhibits endometrial hyperplasia and prevents uterine smooth muscle contraction by antagonizing estrogen, playing a role in protecting pregnancy; the antagonistic effect weakens in late pregnancy, and estrogen levels increase, which is conducive to the initiation of labor and makes it easier for mothers to give birth. The author reviewed the published literature, clarified the regulatory role of estrogen and its nuclear and membrane-related receptors in maintaining endometrial function, and provided a reference for protecting female reproduction. In addition, the author should also describe the effects of drugs such as tamoxifen, raloxifene, fluvastatin and G-15 on the endometrium. Future research should focus on evaluating new therapeutic strategies that precisely target specific ER and its associated growth factor signaling pathways. Progesterone is a synthetic drug similar to progesterone, and 50 years of clinical practice has shown that oral progesterone can effectively treat endometriosis. It is reported that 90% of patients experience pain relief or disappearance after using progesterone. But the problem we face is recurrence after discontinuation of medication. Regardless of the treatment regimen, it is a long-term and repeated drug. During treatment, clinicians need to evaluate not only the efficacy, but also the drug tolerance and treatment costs. Therefore, first-line drugs should be able to be used for a long time with minimal side effects. Progesterone preparations include oral drugs, injectable drugs, subcutaneous implants, and intrauterine devices. Nonsteroidal anti-inflammatory drugs can be used to control pain caused by endometriosis, and those who fail treatment can be given hormone suppression therapy. The main treatment principle is to induce amenorrhea. First-line treatment includes oral contraceptives and oral progesterone, followed by gonadotropin-releasing hormone analogs. Danazol has also been shown to be effective in controlling pain, and intrauterine devices containing levonorgestrel (meniscus) can be used to treat dysmenorrhea caused by endometriosis. Aromatase inhibitors such as letrozole and anastrozole are also effective. The specific mechanism by which progesterone relieves pain in endometriosis is still unclear. Whether related preparations of progesterone can be supplemented, as well as their therapeutic effects and side effects. Further exploration of the medication cycle, dosage and effectiveness of combined treatment is needed.
Comments on the Quality of English Language
Needs minor editing for English
Author Response
We thank the reviewer of his/her interest in our review and his/her enthusiastic comments, especially regarding the part on the mechanism of action of estrogen and progesterone. The reviewer emphasizes the role of progesterone in several biological processes, as well as the use of progestogens or estrogen antagonists.
He is also suggesting addressing two additional issues: 1) the effect of drugs such as tamoxifen, raloxifene, fluvastatin and G-15 on the endometrium, 2) further exploration of the medication cycle, dosage and effectiveness of combined treatment. The reviewer should know that in a previous submission of the manuscript, we addressed these issues. However, the comments of another reviewer and of the editor have suggested to restrain the focus of the review. Therefore, we took the decision to focus our manuscript on the part dedicated to endometrium biology and the signaling of ER and PR. Therefore, these two issues could be discussed in another reviewer more specifically dedicated to the effect of drugs related to estrogens and progesterone.
Reviewer 2 Report (Previous Reviewer 2)
Comments and Suggestions for Authors
Isabelle Dias Da Silva et al. in “Unraveling the Dynamics of Estrogen and Progesterone Signaling in the Endometrium: an Overview” show in extremely depth how the complexity of estrogen and progesterone receptor signaling, which will allow the identification of key pathways that could open new perspectives for the development of drugs that could prevent or treat medical issues related to the endometrium.
I consider original the proposal of organoids and organ-on-a-chip models that will allow the possibility to study crucial steps such as endometrial proliferation, decidualization, embryo-endometrial interface and thus embryo implantation. These models will usefull to evaluate new drugs on human tissue before clinical trials.
The references are appropriate and recent. They support the conceptualizations present in the review.
To improve paragraph 3.2.4. Key Mediators of PR Signaling in Endometrial (Line 547) the authors should add a table with all Key Mediators of PR Signaling.
Comments on the Quality of English Language
Minor editing of English language required
Author Response
We agree with the reviewer that a summary of the key mediators of PR signaling will facilitate the comprehension. Thus, to emphasize the links between these key mediators of PR signaling, instead of presenting a simple table, we completed figure 4 with the key mediators we are reporting throughout the manuscript. We hope that this schematic summary will help the reader and meet the reviewer comment.
Reviewer 3 Report (Previous Reviewer 3)
Comments and Suggestions for Authors
The revised manuscript ‘ Unraveling the Dynamics of Estrogen and Progesterone Signal-2 ing in the Endometrium: an Overview’ was reviewed with interest.
Please find below comments while re-reading the manuscript
Abstract: do not speculate such as ‘ Understanding the complexity of estrogen and progesterone re-16 ceptor signaling will allow the identification of key pathways that could open new perspectives for 17 the development of drugs that could prevent or treat medical issues related to the endometrium’
Introduction
- Fig 1: do women have less estrogens at age 20 than age 30 ?
- Par 2 and 3 are out of scope of the endometrium e.g. pathology as endometriosis is a different discussion
- L 59 : I do not see the relationship of the ‘hypothalamic-pituitary-ovarian (HPO) axis [26]’ for the endometrium
- L63-70 ?? menopause is a different issue
Physiology
- L 94 MHT is out of scope especially with 405 references
- L102 what is a ‘sophisticated stem cells’?
- L 118-119 ‘when estrogens predominate, especially in the 118 basal layer of the endometrium, as opposed to the functional layer intended for monthly 119 shedding [60,74]’ I do not understand
- L158-159 ‘This nu-158 anced understanding of epithelial and stromal cells heterogeneity but also the identifica-159 tion of previously overlooked subpopulations provides a deeper insight into uterine biol-160 ogy during the menstrual cycle.’ The message is the importance of cellular heterogeneity, as demonstrated by different markers. I doubt that listing all of them helps the reader. The conclusion is speculation.
- Implantation, pregnancy and menopause are out of scope
- Organoids is a different discussion
Conclusion
I leave the decision to the editor. The manuscript is very detailed but close to unreadable. My opinion is that a more focused manuscript detailing what is known about estrogen and progesterone effects and their interplay in the endometrium would be preferable
Comments on the Quality of English Language
Overall ok
Author Response
Abstract: do not speculate such as ‘ Understanding the complexity of estrogen and progesterone re-16 ceptor signaling will allow the identification of key pathways that could open new perspectives for 17 the development of drugs that could prevent or treat medical issues related to the endometrium’
To follow the reviewer’s comment, we replace the speculative sentence by this one: Understanding the complexity of estrogen and progesterone receptor signaling will help elucidate the mechanisms underlying normal reproductive physiology and provide fundamental knowledge contributing to a better understanding of the consequences of hormonal imbalances on gynecological conditions and tumorigenesis.
Introduction
- Fig 1: do women have less estrogens at age 20 than age 30 ?
No indeed, there is no differences and we adapted fig 1 accordingly.
- Par 2 and 3 are out of scope of the endometrium e.g. pathology as endometriosis is a different discussion
In this introduction, our goal is to present the importance of studying estrogen and progesterone signaling, which is crucial not only for endometrium biology and function but also implicated in pathological conditions. So we added the following sentences to clarify:
Line 48: Imbalances or disruptions in estrogen and progesterone signaling can lead to various gynecological conditions related to the endometrium affecting fertility and reproductive health. Endometriosis…
Line 58: Studying estrogen and progesterone pathways will bring basis for the management of these conditions, identifying potential targets.
- L 59 : I do not see the relationship of the ‘hypothalamic-pituitary-ovarian (HPO) axis [26]’ for the endometrium
The sentence has been corrected as followed: These processes are modulated by estrogen and progesterone, which are governed by the hypothalamic-pituitary-ovarian (HPO) axis along woman’s reproductive life.
- L63-70 ?? menopause is a different issue
Menopause is a different issue, but it is closely related to fluctuations in hormonal levels impacting endometrium. Thus, understanding estrogen and progesterone signaling in the endometrium is essential to optimize menopause hormone therapies, minimizing risks and maximizing benefits. We thus shortened the paragraph and specified the interest of studying estrogen and progesterone signaling in the endometrium in this context.
Line 64: Nevertheless, the endometrium is also a matter of concern when menopause occurs, as menopause results from the ovaries ceasing to produce estrogen.
Line 70: Thus, understanding estrogen and progesterone signaling in the endometrium is essential to optimize menopause hormone therapies, minimizing risks and maximizing benefits.
Physiology
- L 94 MHT is out of scope especially with 405 references
We agree with the reviewer, and we have remove this chapter from the manuscript.
- L102 what is a ‘sophisticated stem cells’?
This term has been removed from the manuscript. The sentence (line 102) is now: Exploring the intricate functional landscape of the endometrium reveals a rich diversity of cell types derived from stem cells,…
- L 118-119 ‘when estrogens predominate, especially in the 118 basal layer of the endometrium, as opposed to the functional layer intended for monthly 119 shedding [60,74]’ I do not understand
To clarify, we modify the sentence as followed (line 118): This pattern is consistent with the high activity of the basal layer cells during the proliferative phase of the menstrual cycle, when estrogens predominate.
- L158-159 ‘This nu-158 anced understanding of epithelial and stromal cells heterogeneity but also the identifica-159 tion of previously overlooked subpopulations provides a deeper insight into uterine biol-160 ogy during the menstrual cycle.’ The message is the importance of cellular heterogeneity, as demonstrated by different markers. I doubt that listing all of them helps the reader. The conclusion is speculation.
We modified the conclusion as followed (line 156): This study highlights the heterogeneity of stromal cells and identifies previously overlooked subpopulations that are important during the menstrual cycle. We think that providing a list of these 4 markers is interesting, especially for researchers that would like to focus on one of these stromal cell population.
- Implantation, pregnancy and menopause are out of scope
We did not discuss in detail the processes of implantation and pregnancy. However, we discussed the crucial mediators involved, especially in PR signaling to undergo decidualization and prepare the endometrium to proper implantation.
We agree with the reviewer that menopause was out of scope and removed this chapter from the manuscript.
- Organoids is a different discussion
We only mentioned organoids as a useful tool. In addition, since another reviewer of the manuscript has considered original the proposal of organoids and organ-on-a-chip models that will allow the possibility to study various aspect of endometrium biology, we decided to keep this idea in the manuscript.
Conclusion
I leave the decision to the editor. The manuscript is very detailed but close to unreadable. My opinion is that a more focused manuscript detailing what is known about estrogen and progesterone effects and their interplay in the endometrium would be preferable.
We adapted the manuscript to facilitate the reading of the text and we hoped we better emphasized the relationship between the molecular mediators and the hormonal effects on endometrium biology. We removed several paragraphs, and we clarified a lot of sentences that are highlighted in red. Moreover, we added subtitle sections (lines 319, 367, 512, 550, 525) to better emphasize the relationship or interplay between estrogen or progesterone effects, and the mediators implicated in ER or PR signaling.
For the Key Mediators of ERα Signaling in the Endometrium, we specified (line 315) that ERα Signaling is mainly implicated in the proliferation of the epithelial cells of the en-dometrium occurring during the proliferative phase of the menstrual cycle, and in the preparation to decidualization and proper receptivity during the secretory phase. Then we subdivided this section between the following sub-sections: 1) Proliferation-Related Mediators (line 319), and 2) Mediators Involved in the Preparation to Decidualization and Implantation (line 367).
For the Key Mediators of PR Signaling in the Endometrium, we specified (Line 506) that P4/PR signaling inhibits the proliferative effect of E2 observed in epithelial cells and controls the decidualization of stromal cells. In addition, P4/PR signaling drives the differentiation of luminal epithelial cells into ciliated cells, and of glandular epithelial cells into secretory cells. Numerous mediators and molecular modulators, essential for de-cidualization and receptivity, depend on P4/PR signaling and are involved in epitheli-al-stromal interactions [260] (Figure 4). Then we subdivided this section between the following sub-sections: 1) Mediators Counteracting ER Signaling (line 512), 2) Main Mediators of Decidualization (line 550), and 3) Other Partners of Decidualization (line 525).

This manuscript is a resubmission of an earlier submission. The following is a list of the peer review reports and author responses from that submission.
Round 1
Reviewer 1 Report
Comments and Suggestions for Authors
This is a very comprehensive review introducing the effects of estrogen and progesterone on the endometrium. It provides an almost comprehensive introduction to the research progress in physiology and drug interference in this field. During menopause, progesterone occurs before estrogen deficiency. In other words, the endometrium continues to proliferate and thicken under the stimulation of estrogen, but lacks sufficient progesterone to protect it. At this point, it is easy to develop endometrial hyperplasia, lesions, and even endometrial cancer. This article also mainly introduces two hormone related drugs for the treatment of endometrial related diseases. I can no longer better describe the beauty of this article. This review will provide us with good reference for future research and clinical treatment of endometrial related pathology and physiology. During menopause and after menopause, there is a significant decrease in estrogen in women's bodies, while the decrease in androgens is much slower with age. The author can add some progress on testosterone in the endometrium in the outlook section.
Comments on the Quality of English Language
Moderate editing of English language required
Reviewer 2 Report
Comments and Suggestions for Authors
Comments and Suggestions for Authors Isabelle Dias Da Silva et al. in “Hormone therapy in menopause and the endometrium: unraveling the dynamics of estrogen and progesterone signaling” shows the physiology of the endometrium, understanding the complex signaling pathways of estrogen and progesterone, before focusing on recent advances in the development of new MHTs, such as tibolone and TSEC based on estrogen/BZA combination such as CEE/BZA, which may offer new opportunities for patients to improve quality of life without posing a threat to the breast and endometrium.
The review is well articulated and the topics are in-depth, but the authors should better explain the relationship between MHT and Endometrium line 761.
Comments on the Quality of English Language
Minor editing of English language required
Reviewer 3 Report
Comments and Suggestions for Authors
The manuscript ‘ Menopausal Hormone Therapy and Endometrium: Unraveling the Dynamics of Estrogen and Progesterone Signaling’ has been reviewed with interest. The manuscript is an extensive compilation with over 450 references. Despite its merits, the manuscript is not suited for publication in its present form. The main problem is the overall feeling of a selective compilation, an unclear message, and an incomplete and potentially biased extension to HRT.
The mechanism of action of estrogens and progesterone involved seems well described. However, the clinician involved with HRT does not find answers to many of his questions, such as:
- It is nice to know several receptors, including a membrane receptor, exist. However, after reading the article, the pathophysiologies of these receptors are not clear, such as differences during the cycle between epithelial and stromal cells, between the different types of estrogens, and of progestins.
- The abstract raises the expectation of products safe for the breast and the endometrium. However, it remains unclear whether the mechanisms in the breast and endometrium are similar. Data do not explain why unopposed estrogen therapy might induce endometrial cancers, and the extension to basodoxifene reads as potentially speculative and selective.
- Whether progesterone or not testo derivatives have similar or different effects remains unclear.
In conclusion, this manuscript lacks focus and a message
- A review of the mechanism of action and the different pathways of estrogens and progesterone is nice.
- A review of the mechanism of action of the different estrogens, progestins and HRT would be nice
- (Similarly a review of the mechanisms of action on other tissues as endometriosis would be nice)
- A discussion of the advantages and disadvantages of HRT is important
- However, mixing mechanisms of action and HRT needs focus and a message.
Round 2
Reviewer 3 Report
Comments and Suggestions for Authors
The revised version of ‘Menopausal Hormone Therapy and Endometrium: Unraveling the Dynamics of Estrogen and Progesterone Signaling’ has been reviewed and the efforts of the authors are are appreciated. Unfortunately the misunderstanding persists. Therefore , the handling editor should decide, to avoid a polemic.
I try to help; not to criticise. This manuscripts deals with 3 different topics
The part on the mechanism of action of estrogens and progesterone is nice and informative.
The second part deals with the pharmacology of the different estrogenic and progestagenic compounds. Unfortunately, there is little relationship with understanding the mechanism of action since our understanding is poor and the description is rather eclectic. E.g. why are progestins so different in ovulation inhibition effect and clauberg effect?; are differences in effect of estrogens and in progestins more than differences in receptor afficinity?; what is the mechanism of the androgenic or anti-androgenic or anti mineralocorticoid effects of the progestins?
The third part deals with clinical aspects of hormone replacement therapy (called MHT) such as breast cancer and endometrial cancer risk and prevention. After reading the manuscript, the clinician will not have an answer to the many questions. For breast and endometrial cancer: Do estrogens increase the risk or only the growth?; is an effect concentration-dependent or a consequence of bioavailability? Any difference between progestins? Etc.
Answer:
We thank the reviewer of his/her interest in our review especially regarding the part on the
mechanism of action of estrogen and progesterone. We regret that despite our e:orts to
address his/her comments from the round 1, we were not able to su:iciently clarify the
message of our manuscript. Thus, in an attempt to address his/her comment, especially
regarding the lack of focus, we took the decision to focus our manuscript on the first part
dedicated to endometrium biology and the signaling of ER and PR. The title of the
manuscript has been adapted accordingly.
In addition, to meet the Editor’s recommendation, we modify the introduction section in
order to emphasize the general and multiple clinical contexts related to the endometrium.
For your convenience, the modifications from the previous submission are highlighted in
red throughout the manuscript.